# A Simple Model of Knowledge Scaffolding Applied to Wikipedia Growth

**Franco Bagnoli** [1,2,*,†] and **Guido de Bonfioli Cavalcabo'** [1,†]

1 Department of Physics and Astronomy and CSDC, University of Florence, Via G. Sansone 1, 50019 Sesto Fiorentino, Italy
2 INFN, sez. Firenze, Via G. Sansone 1, 50019 Sesto Fiorentino, Italy
* Correspondence: franco.bagnoli@unifi.it
† These authors contributed equally to this work.

**Abstract:** We illustrate a simple model of knowledge scaffolding, based on the process of building a corpus of knowledge, each item of which is linked to "previous" ones. The basic idea is that the relationships among the items of corpus can be essentially drawn as an acyclic network, in which topmost contributions are "derived" from items at lower levels. When a new item is added to the corpus, we impose a limit to the maximum unit increase (i.e., "jumps") of knowledge. We analyzed the time growth of the corpus (number of items) and the maximum knowledge, both showing a power law. Another result was that the number of "holes" in the knowledge corpus always remains limited. Our model can be used as a rough approximation to the asymptotic growth of Wikipedia, and indeed, actual data show a certain resemblance with our model. Assuming that the user base is growing, at beginning, in an exponential way, one can also recover the early phases of Wikipedia growth.

**Keywords:** Wikipedia; knowledge organization; scaffolding model; knowledge modelling; knowledge visualization; knowledge graph

## 1. Introduction—Knowledge Scaffolding

Knowledge is created by the human brain, and then it is amplified and integrated into organizational knowledge by social interactions. We wanted to develop a simple model of such knowledge building.

During the last century, it was widely observed that human society had been gradually turning into a "knowledge society" [1]. Thanks to the evolution of the communication technology and the diffusion of the World Wide Web, there are almost no limits on when, where and how knowledge can be transferred among individuals, and that is why knowledge flows have become more intersectoral, interorganizational, interdisciplinary and international [2].

Collecting data and information has an increasingly central role in almost any process involved in the development and progress of a society. Knowledge graphs are used to put into context this collection of interlinked descriptions of concepts, occurrences, objects and relationships. The investigation of knowledge graphs [3] aims to derive new conclusions from existing data and possible errors in previous analysis.

In all scientific disciplines based on deduction, such as most of mathematics and in many parts of physics, the derivation of higher-level achievements depends on previous knowledge. In these cases, new pieces of knowledge that are accepted and inserted into the existing corpus are based on previous results. Therefore, it is not correct to define this process by the word "accumulation"; instead, "scaffolding" [4] is the best-suited term in our opinion, since knowledge is not gathered into a disordered pile, but scaffolded into an ordered pattern that gives a structure to human activity by supporting and guiding it [5].

This process corresponds to an acyclic direct network [6], in which the nodes are the bits of knowledge and the links represent connections between each new item and its prerequisites; i.e., the elements of the existing corpus needed to prove the new result. In acyclic graphs the nodes are not mutually interlinked with each other, and therefore, the network does not contain any cycle. This means that no path ever returns to a visited link.

Let us illustrate the toy model shown in Figure 1. We have a lattice of a certain width *W* that grows from the bottom. Each node represents an item of knowledge and can be "empty" (black) or inserted in the corpus (red or green circles). The first row, in green, represents the axioms, i.e., the starting knowledge corpus. At each time step, an empty node whose height is smaller or equal to the maximum height plus the range *L* is chosen at random. This node chooses *K* other nodes, with lower height. If any of these nodes are still empty, the new node is discarded; otherwise, it is inserted in the corpus, and the maximum height is eventually updated. In this model, the levels are clearly visible, although the width *W* is an arbitrary parameter. In principle, one should use a variable-width model in which the structure becomes evident only at the end.

In the snapshot of Figure 1 right, "knowledge holes", i.e., missing items at levels which are lower than the highest one, and that will be subsequently filled, are clearly shown. Notice that holes are evident in this fixed-width version, but can be identified also in the variable-width model by observing the temporal filling of the structure, i.e., the appearance of items below the maximum ones. We revise the properties of our actual model, which were already presented in Ref. [7], in Section 4.

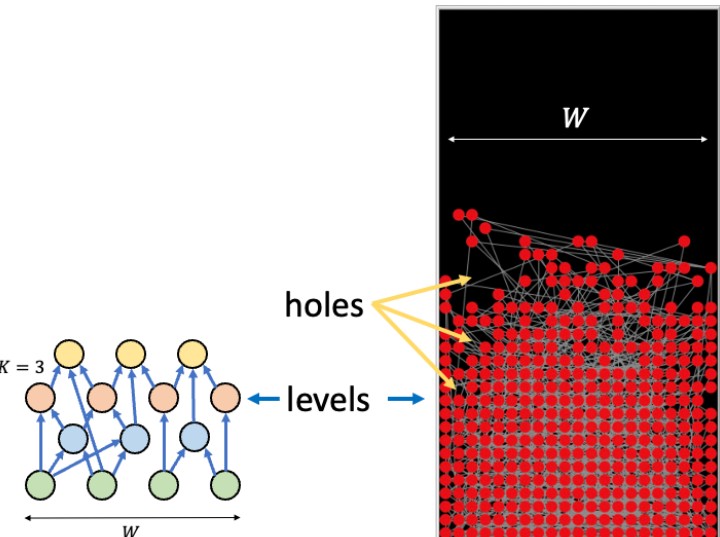

**Figure 1.** The fundamental knowledge scaffolding model. (**left**) Knowledge bits are represented as nodes of a network, where different colors represent different levels and nodes at a certain level only depend on a certain number of nodes at lower levels. Green (basic) nodes represent axioms. (**right**) Observing the filling of the network (here with fixed width *W* and with fixed number of dependencies *K*), one can detect holes that are filled after the appearance of nodes at higher levels.

## 2. Introduction—Application to Wikipedia

Although the original model was developed while thinking of a structured knowledge corpus, it can be used also to interpret less organized structure. Among the platforms where knowledge is accessible online, Wikipedia, the multilingual, free, online encyclopedia written and maintained by a community of volunteers, called wikipedians, through open collaboration, is the most famous one [8].

Wikipedia has become the only nonprofit venture among the Internet's top 100 most popular websites [9] and an important resource for many students, but also for numerous experts, even at an academic level [10]. Since 2001, Wikipedia's number of pages has grown every year.

The number of pages can be roughly derived from two factors: the number of users and the average number of pages produced by each user, in the unit of time (say, one year). Indeed, Wikipedia has witnessed quick growth of the number of users, to which recently bots (i.e., algorithms) where also added. However, it seems that both the growth of the number of users and their activity have declined in time [11].

The number of active users exhibits an initial exponentially-like growth followed by a stabilization phase [12]. However, Wikipedia does not distinguish between the creation of new pages and updates. Moreover, a large number of edits are performed by anonymous users.

From another point of view, the first growth of Wikipedia was mainly fed by "porting" existing pieces of knowledge from other sources. It was followed by a "refining" phase, in which long articles were split in pieces, following the recommendations in [13]. This phase may be considered part of the hole-filling process. Finally, it is expected that asymptotically, the user base will reach a stable size, and that only new knowledge will be added, which is the main ingredient of our model.

However, we also have to consider that Wikipedia articles, differently from articles published in journals, are never finished, being continuously polished and edited [14].

Wikipedia certainly does not have an acyclic character, not even in the parts referring to mathematical concepts (as do other scientific libraries), except maybe for the "see also" links [15]. However, the category structure is indeed almost acyclic, indicating a directed logical character at a higher level [16].

We only analyzed the English language part of the website because, with almost 11% of the total articles available, it is the largest portion [17]. The total number of pages of the English language portion has almost reached 57 million, of which articles make up 11.6% [18], and that is just one of the 318 currently active editions of the online encyclopedia.

In the following, we describe a simple and unstructured model representing the linear progress of the growth of knowledge as a linear scaffolding. The study of the knowledge growth in our model could be seen as an "ex-post" analysis of the temporal development of the considered knowledge space. In the same way, the analysis of the growth of the number of pages in Wikipedia can be seen as a simple way to study the dynamic of global knowledge.

This is clearly a simplification; adding content to an existing page or removing content previously considered correct are aspects that our model does not take into account. However, just as we sought simplicity in our model, we seek simple modeling to describe articulated systems, such as the evolution of the global knowledge in Wikipedia. Additionally, the fact that we are talking of more than 6.5 million articles facilitates the construction of a model that highlights a trend.

In summary, in this paper we want to investigate if our model of "linear" knowledge scaffolding is able to provide an approximation of the growth in articles in the English part of Wikipedia.

Our main research question, however, was not that of fitting data, but rather to develop a plausible agent-base model [19] that "builds" the system of knowledge scaffolding and creates a corpus of known items.

The principal threat to validity of our approach is the usual one shared with other "constructive" models: the agents that represent real users are extremely simplified when trying to capture the essential "ingredients" of a complex system such as Wikipedia. In the present approach, we disregard many aspects, such as motivations, interactions among users (e.g., discussions), editing, improvements to pages and so on.

Some related works on Wikipedia's growth are presented in the following section. The model is presented in a detailed way in Section 4. Numerical and analytical results are reported in Section 5. Comparison with Wikipedia's actual growth is presented in Section 6, and the model is extended to include a growing number of users in Section 7. Discussions and conclusions are drawn in Section 8.

## 3. Related Work

Papers about Wikipedia's growth are mostly concerned with data analyses, trying to figure out the best mathematical description for the growing trends [20].

The English Wikipedia has been considered an important resource since shortly after its creation in 2001. By 2005, it was already positively compared with one of the most important English-language encyclopedias [21], and recently it has been pointed out how it leads to changes in the scientific literature as well [22].

Between 2003 and 2006, the general model for the article count of Wikipedia was an exponential model [20]. In this first phase, the growth is dominated by the "importing" of prior knowledge that is easily accessible from other sources, a phase governed by a self-similar process that has a more than exponential trend because it also coincides with the growth in the number of contributors.

At that time, the growth of content creation attracted more users, who in turn created more content, though there was no increase in the contributors' productivity; it actually decreased [23]. This phase was modeled as a preferential attachment process [24] and was definitely overcome around the end of 2006 when the rate of growth peaked and began declining [25].

The next phase of Wikipedia growth showed a slowdown, because the number of contributors reached the physiological threshold. Additionally, the creation of articles from already available data slowed down, since many of the known facts were already present on the website. In this phase, even the editing activity suggests that Wikipedia's growth slowed [11]. The exponential trend at that point has been analyzed again and described as the early part of a logistic curve. However, as discussed below, we do not expect real logistic growth, but rather a switch to a slower pace.

Although discussions regarding the many articles published in Wikipedia have increased [14], the growth of the number of articles itself slowed down drastically because of our finite amount of encyclopedic knowledge [26]. Nevertheless, Wikipedia's growth was still reproduced with excellent agreement using a different model of preferential attachment [27,28].

Finally, once most knowledge is cataloged on Wikipedia, we will have an even slower growth phase due mainly to the entry of new knowledge items that will be discovered "inside" or "outside" the knowledge corpus of the time.

This happens because the knowledge network of Wikipedia not only grows bigger but it also "expands" inward by filling knowledge holes (gaps). According to the literature, the process of gap formation and filling should be valued by the scientific community, since it produces discoveries that are more often awarded Nobel prizes than other processes [29].

It was also noted that Wikipedia, after the initial rapid expansion, is more effective at quantifying knowledge growth than other knowledge networks, such as journals [30].

The model presented in the next section tries to mimic how new knowledge is acquired, and therefore, this asymptotic growth rate of Wikipedia.

## 4. The Model

Since a graphical representation of knowledge acquisition is given by an oriented or growing network [31], we could represent knowledge scaffolding as shown in Figure 1. This feed-forward structure can be seen as a layered network where an item at any level depends on other items at lower levels.

We chose instead a simpler, linear progression, as shown in Figure 2. This is of course a simplification of the process, because, for example, with a linear 1D model, we cannot put into evidence the structures that group together all simultaneous and independent derivations.

The analysis in this case is easier, and it should not impair the conceptual results. As we can consider all items with the same highest prerequisite as simultaneous, we privileged the search for simplicity in order to put into evidence the plausible essential

components of an observed phenomenon and to avoid too many variables that could be tweaked to fit any data set available.

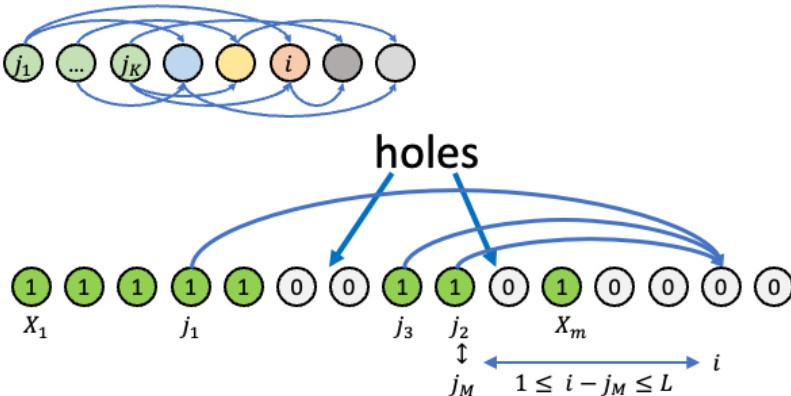

**Figure 2.** The linear scaffolding model. In this version, there are no well-defined levels. Known items are denoted by ones in a linear array of elements (*X*), and unknown ones by zeros. The "highest" known item is at position *m*. New items (*i*) become known ($X_i = 1$) if all random prerequisites $j_k$ are known. The maximum knowledge jump *L* limits the choice of *i* so that $1 \leq i - j_M \leq L$, where $j_M$ is the highest of prerequisites $j_k$. In the figure, the proposed new item *i* is based on prerequisites $j_1$, $j_2 = J_M$, and $j_3$ (*K* = 3).

The knowledge space is represented by an array $X_i$ of *N* items, which can be known ($X_i = 1$) or unknown ($X_i = 0$). We denote by

$$c(t) = \sum_i X_i \tag{1}$$

the size of the knowledge corpus at a certain time.

The linear scaffolding process can be modeled by a random proposal of contributions, each of which depends on a certain number *K* of prerequisites, and provide a higher contribution. We assume that there is a limit to jumps in knowledge, and therefore, the newly proposed item cannot be at a distance greater than *L* from the highest prerequisite (Figure 2).

In the structure of our one-dimensional model, the distance between any two nodes can be roughly seen as the difference in prerequisites, or already known items of knowledge, needed to obtain the considered item, and *L* can be seen as the higher limit of how many consecutive knowledge items could be obtained using only the known items up to a certain point.

Let us denote by *m* the index of the highest known item in the corpus, i.e., $X_m = 1$ and $X_j = 0 \; \forall j > m$. The corpus can contain holes, i.e., $X_j = 0$ with $j < m$.

A contribution cannot refer to prerequisites higher than *m*, but could be rejected because it is based on holes in the present corpus. Redundant contributions (i.e., articles providing already known items) are not considered, even if they are valid. These rejected pages are the equivalents of those marked as duplicates (and subsequently merged) in Wikipedia.

The corpus filling proceeds by randomly choosing *K* items, $j_1, \ldots, j_k, \ldots, j_K$, with $j_k \leq m$, as the prerequisite for a new contribution. It may happen that two or more of the $j_k$ correspond to the same item, since they are chosen at random in the interval $0, \ldots, m$.

We denote by $j_M$ the largest values of the selected $j_k$, and we extract a random integer *i* such that

$$i \in (j_M + 1, \ldots, j_M + L) \tag{2}$$

as a candidate for the new piece of knowledge.

If all the prerequisites are known (i.e., $X_{j_k} = 1 \, \forall k$) and the piece of knowledge $X_i$ is not already known (i.e., $X_i = 0$), then this contribution is added to the corpus ($X_i = 1$ and $c$ is incremented by one), and if $i > m$, then $m = i$. In any case, the time $t$ is incremented by one.

At the beginning, we start with a knowledge vector of zeros, except the $2K$ smallest locations, that represent the starting points ("axioms") from which the knowledge structure is built.

The choice of the number of axioms is arbitrary, and in our case $2K$ was chosen so that the first contribution is not forced to use all present axioms (since each item is based on $K$ prerequisites), but the number of starting axioms does not influence the evolution of the corpus.

For $L > 1$, the knowledge corpus may contain holes, i.e., locations $\ell$ with $\ell < m$ and $X_\ell = 0$. We denote by $h(t)$ the number of holes in the corpus at time $t$.

We repeat the previous steps until the maximum knowledge $m$ is equal to $N$.

## 5. Simulation Results

Typical graphs for the maximum knowledge $m(t)$ and corpus size $c(t)$ are reported in Figure 3, where we can also see that both values grow with the same trend regardless of the values of $K$ and $L$, separated by a gap that remains finite after an initial growth phase. This gap is related to the number of holes in the knowledge corpus.

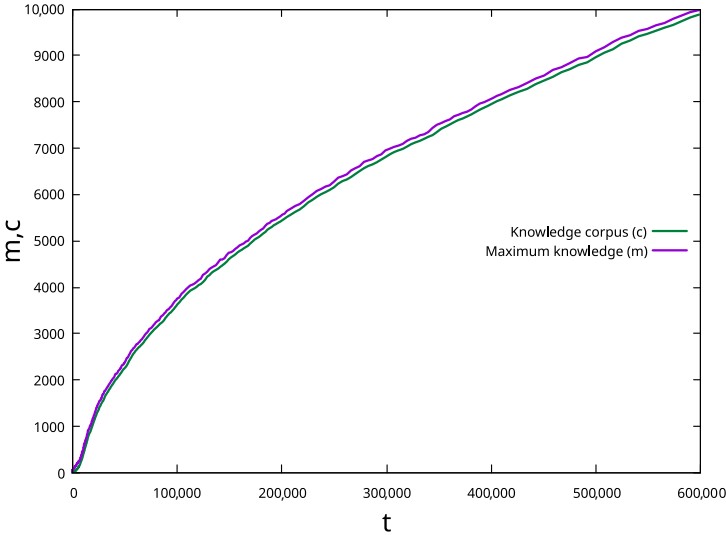

**Figure 3.** Comparison , between the evolution of the knowledge corpus ($c$) and maximum knowledge ($m$) vs. time for $N = 10,000$, $K = 4$ and $L = 20$.

By plotting $c(t)$ in a log–log scale, as shown in Figure 4, we noticed that, regardless of the values chosen for $L$ and $K$, it is asymptotically approximated by a power law:

$$m(t) = \mu(K, L)t^\alpha \tag{3}$$

and

$$c(t) = \gamma(K, L)t^\alpha, \tag{4}$$

with $\alpha = 1/2$.

Looking at Figure 4, we can notice that the transient depends on $K$ and $L$: ny increasing these values, the curve becomes less linear in the left part of the graph. However, the asymptotic trend is independent of $K$ and $L$, as shown in Figure 5.

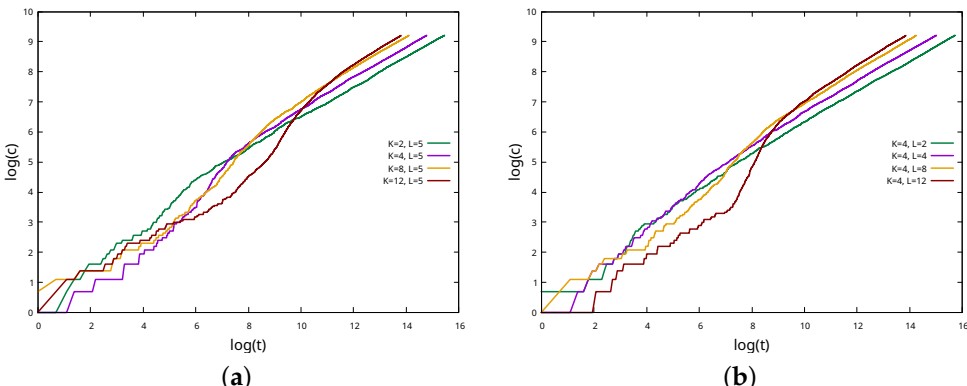

**Figure 4.** Logarithmic plot of the knowledge corpus *c* for $N = 10{,}000$. (**a**) Fixed $L = 5$ and five values of *K*. (**b**) Fixed $K = 4$ and five values of *L*.

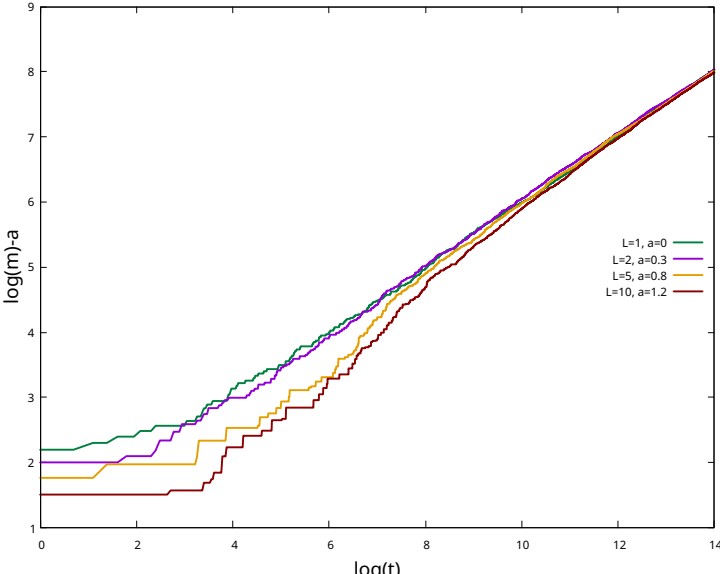

**Figure 5.** Logarithmic plot of knowledge corpus *c* for $N = 10{,}000$ and $K = 4$. It is possible to numerically rescale the function with different *L* to obtain an asymptotic convergence. The same results could be obtained by varying *K*.

By investigating the number of holes $h(t)$, we can see that this value remains quite constant and that this effect is present regardless of the values of *L* and *K*, as reported in Figure 6. Indeed, *h* increases at first, and the value around which the number of holes oscillates increases with *L*, and in a less evident way, also with of *K*, but eventually, for every *K* and *L*, the the filling of inner holes takes place, in correspondence with a slowing growth of $c(t)$.

Let us consider the simplest case possible: $K = L = 1$, for which there is no hole in the corpus, since every piece of knowledge depends on the immediately previous one. In this case, $c = m$.

For each time step, either the knowledge increases by one, or it stays constant. It is thus an example of a birth-death Markov Chain, with no deaths [32].

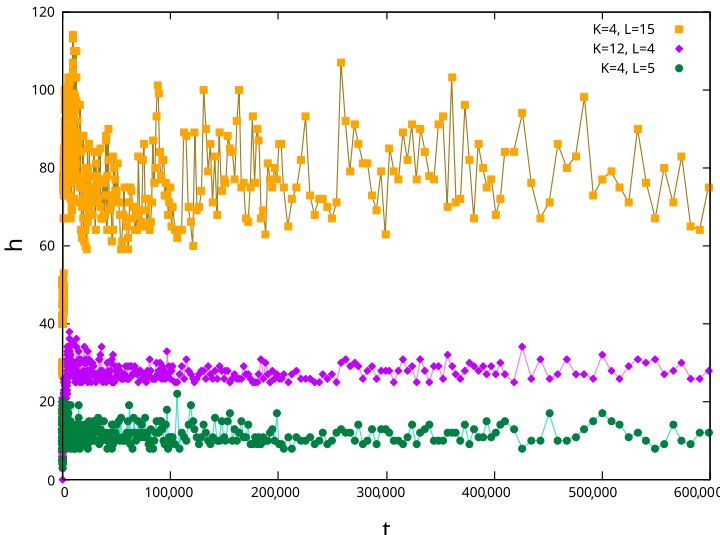

**Figure 6.** Time evolution of the number of holes ($h$) for $N = 10{,}000$ and different values of $K$ and $L$.

Let us denote by $P(y, t)$ the probability that at time $t$ the corpus size and the maximum knowledge level are both $y$. The evolution of $P$ is given by

$$P(y, t+1) = P(y, t) \left( \frac{y-1}{y} \right) + P(y-1, t) \left( \frac{1}{y-1} \right), \tag{5}$$

where there are two processes: the knowledge increases by one if the $j_1$ prerequisite is equal to $y - 1$ (with probability $1/(y-1)$), or stays constant and equal to $y$ if $j_1$ is one of the other $y - 1$ possibilities.

The average knowledge $\bar{y}(t)$ at time $t$ is defined as

$$\bar{y}(t) = \sum_y y P(y, t). \tag{6}$$

The Markov process starts from the condition $P(y, 0) = \delta_{y, 2K}$.

The time and space continuous approximation of the Markov Equation (5) (valid for $y \gg 2K$ and $t \gg 0$) is:

$$\frac{\partial P}{\partial t} = -\frac{1}{y} \frac{\partial P}{\partial y}, \tag{7}$$

which implies that $P$ is a function of $y^2 - 2t$. The distribution $P(t)$ always keeps a sharp-peaked shape, implying that the average value of $y$, which corresponds to the average maximum knowledge $m(t)$, follows the same time behavior of a generic $y$.

In this approximation,

$$\bar{y}(t) = m(t) \propto \sqrt{2t} \tag{8}$$

for large $t$.

The approximation Equation (5) was obtained for the case $L = 1$, $K = 1$. It is possible to generalize it to the case $L > 1$, imposing the absence of holes, as

$$P(y, t+1) = \frac{1}{Z(t)} \sum_{k=0}^{L} P(y-k-1, t) \left( \frac{L-k+1}{L(y-k-1)} \right), \tag{9}$$

where $y - k - 1 > 0$ and $Z$ is a normalization constant, such that $\sum_y P(y, t) = 1$. This generalization is justified by the fact that even for $L > 1$, the number of holes stays limited. This approximation reproduces the power-law growth of the corpus shown by numerical simulations.

The origin of the power law growth is that the corpus either grows or stays the same, but the probability of increasing decreases with the size of the corpus, which is essentially the meaning of Equation (7).

## 6. Comparison with Wikipedia

We are interested in studying the evolution of Wikipedia as a knowledge corpus. As explained in Ref. [18], there are many ways to measure the size of Wikipedia. We used the number of articles (pages that have encyclopedic information on it) as a measure of the corpus size, growing as shown in Figure 7. We considered the number of page edits as an indication of the number of holes of information.

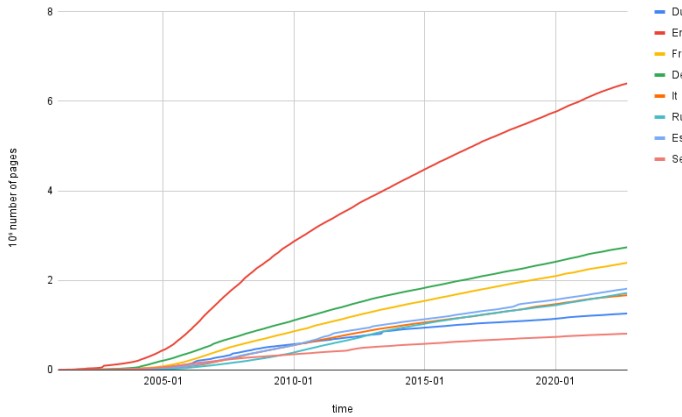

**Figure 7.** Time evolution of number of pages created by users of some of the biggest Wikipedia editions. Codes: Du = Dutch, En = English, Fr = French, De = German, It = Italian, Ru = Russian, Es = Spanish, Se = Swedish. Data collected from [33].

We retrieved the public statistics data of the Wikimedia Foundation for the English Wikipedia from the Wikistats website [34], including data from the foundation of the English Wikipedia in January 2001 to the moment of our acquisition on 1 December 2022.

We have considered only articles and edits created by users, since some Wikipedia editions, such as the Swedish and Dutch ones, have a faster growth rate with a discontinuous trend that is mostly due to bots, such as the Lsjbot [35], as shown in Figure 8. We therefore removed from the counts all pages and edits made by registered bots and users that include "bot" in their username.

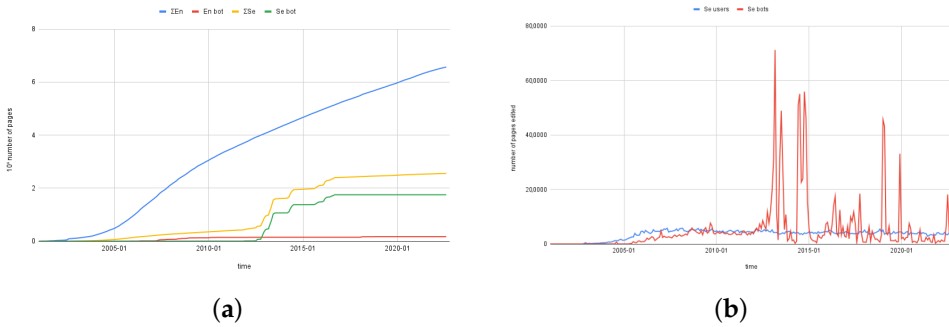

(**a**)                                                                                      (**b**)

**Figure 8.** (**a**) Evolution over time of the total number of articles present in the English (En) and Swedish (Se) Wikipedias and the corresponding bot-created items. (**b**) Number of articles edited, excluding redirect pages, by users and by bots in the Swedish Wikipedia. We can see how the discontinuity in the trend for both counts is due to bot activity. Data collected from [33].

Additionally, as we can see in Figure 9, the growth trends of some of the other large Wikipedias in different languages have approximately the same trend, though lagging

some years behind, and therefore, we can talk about Wikipedia's evolution considering the English trend as the typical one.

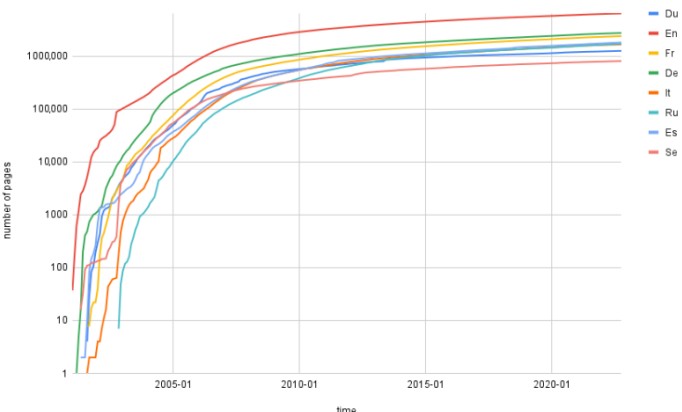

**Figure 9.** Logarithmic trend of the growth of pages created by users of some of the biggest Wikipedia editions. Codes: Du = Dutch, En = English,Fr = French, De = German, It = Italian, Ru = Russian, Es = Spanish, Se = Swedish. Data collected from [33].

The early phases of Wikipedia's growth (2003–2009) can be modeled by an exponential curve, as in Ref. [20], or by a power law with a large exponent, as shown in Figure 10. The latest phase (since 2009) is more coherent with power-law growth with an exponent of less than one, between 0.8 and 0.9. Our mathematical model asymptotically follows power-law growth $m(t) = \mu(K, L)t^\alpha$ with $\alpha = 0.5$

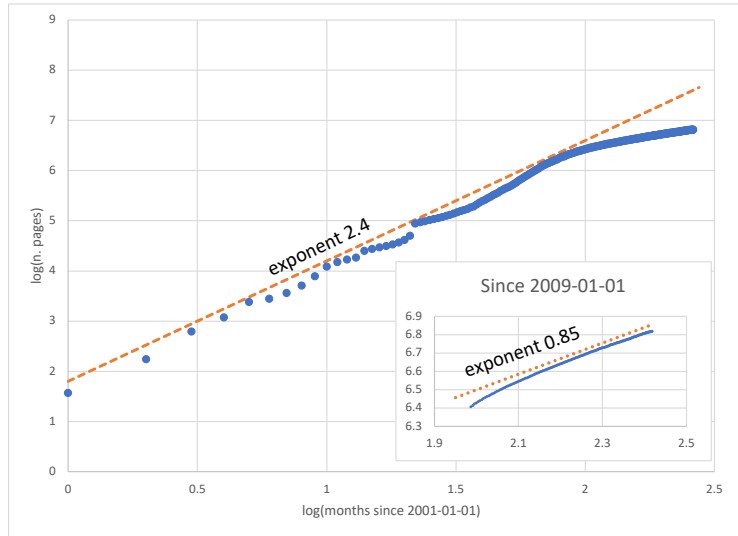

**Figure 10.** Log–log plot of the number of Wikipedia pages from 1 January 2001 (recorded each month). The initial growth can be approximated by a power law with an exponent of about 2.4, but there is a marked change in growth around 1 January 2009, with an exponent of about 0.8 and decreasing.

The basic assumption of our model is a constant contribution of items, which represent the unit of time. In the real world, there are several factors that can affect this rate—first of all the number of users (that at least in the first stages has grown with a large pace), then their average activity and the use of automatic tools (bots), which are not included in our model. In the following section, we examine how a growing user base can be incorporated into our model.

When a hole is present in our model, it means that an "ex-post" analysis tells us that some information was missing until a certain time. By considering the edits as a way to fill

holes in knowledge, we can easily compare the two quantities. As we can see in Figure 11, there is some strong similarity between the number of holes and the fraction of changes made to the pages.

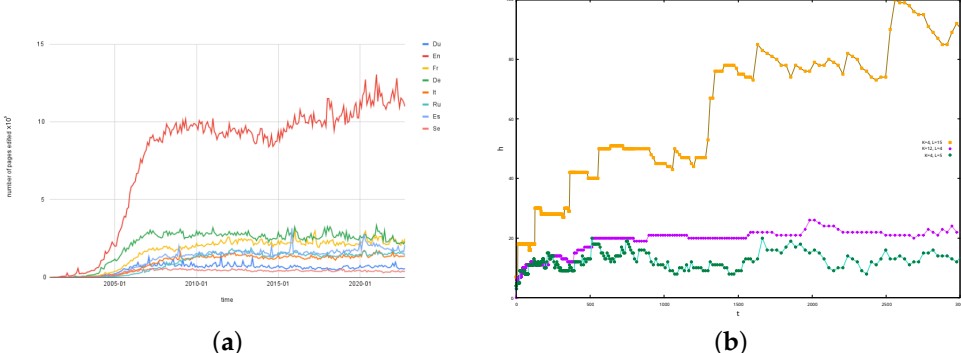

(**a**)                                        (**b**)

**Figure 11.** (**a**) Number of pages edited by users in hundreds of thousands ($10^5$), excluding redirect pages of some of the biggest Wikipedia editions. Codes: Du = Dutch, En = English, Fr = French, De = German, It = Italian, Ru = Russian, Es = Spanish, Se = Swedish. Data collected from [33]. (**b**) Zoom of the first part of Figure 6, where for $L = 15$ the trend has not stabilized yet.

## 7. Growing User Community

The "timing" in our model is on the basis of contributions. However, if the size of the user community grows (or shrinks), the correspondence between "contribution clock" and real time breaks.

Let us denote by $M(t)$ the number of active users, assuming that their average activity is not too dissimilar. In a unit of "real" time there are a number of contributions proportional to $M(t)$, so the time for each contribution is in the order of $1/M(t)$.

In other words, it is possible to rescale our results given the number of users. Let us explore the hypothesis of an initially exponentially growing user base by means of the Markov approximation.

Equation (5) becomes

$$P\left(y, t + \frac{\alpha}{M(t)}\right) = P(y,t)\left(\frac{y-1}{y}\right) + P(y-1,t)\left(\frac{1}{y-1}\right),\tag{10}$$

and its continuous approximation is

$$\frac{\partial P}{\partial t} = -\frac{M(t)}{y}\frac{\partial P}{\partial y}.\tag{11}$$

If $M$ grows exponentially, $\dot{M} = \alpha M$, and we have that $P$ depends on $y^2 - 2\alpha M$. Since $P$, starting from a delta (the initial contributions), always keeps a sharp peaked shape, the average maximum knowledge is

$$m\bar{y}(t) = m(t) \simeq \sqrt{\frac{2}{\alpha}M} = \sqrt{\frac{2}{\alpha}}\exp\left(\frac{\alpha}{2}t\right).\tag{12}$$

It is thus plausible that an exponentially growing user base produces an exponential growing corpus, even in our model.

Let us now check what happens for a logistic growth of the user base, which can be considered as a rough estimation of the actual base consistency [12]. The corresponding logistic equation is

$$\dot{M} = \alpha M(1 - M).\tag{13}$$

We do not have an analytical solution for this case, but the partial differential equation (Equation (11)), which can be discretized as

$$P(y, t+1)) = P(y,t) - \frac{M(t)}{y}\big(P(y,t) - P(y-1,t)\big), \qquad (14)$$

can be numerically integrated.

The growth of the maximum knowledge $m(t)$ is reported in Figure 12 and again has a shape that resembles that of Wikipedia's growth. The first phase is very near to an exponential, as shown in Figure 13a, but asymptotically, when $M(t)$ becomes a constant, it approximates the power-law growth with exponent 1/2; see Figure 13b.

It is not easy to get a correct time estimate of the number of active users to be fed into our model, since Wikipedia records all editing per user regardless of the creation of new pages, editing and discussions, and allows anonymous contributions.

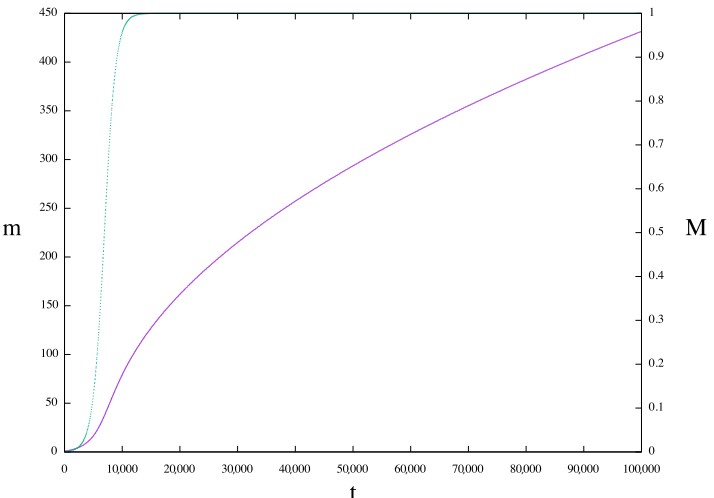

**Figure 12.** Plot of the average value of the maximum height of corpus (approximating the corpus site) $m(t)$ versus time $t$ (continuous line), and the logistic growth of the user base $M(t)$ (dotted line).

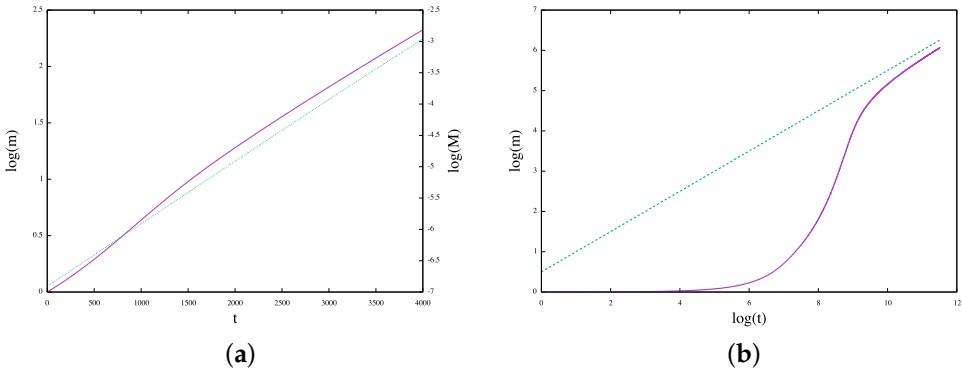

**Figure 13.** (**a**) Log–log plot of the maximum height of corpus (approximating the corpus site) $m(t)$ versus time $t$ for early times. (**b**) The log–log plot of $m(t)$ versus $t$. The dotted line marks the exponent 1/2.

## 8. Discussion and Conclusions

We presented a simple model of knowledge dynamics in which topics are arranged as an acyclic network.

The process of knowledge scaffolding (accumulation) is a random process in which the size of the corpus $c$ and the maximum knowledge $m$ grow with time following a power law with exponent $\alpha = \frac{1}{2}$, once we have fixed the connectivity $K$ and the maximum distance

from the previous knowledge $L$. Asymptotically, we always obtain an approximately constant number of holes. Both those trends are independent of the values of $K$ and $L$.

We have found an analytical approximation of the process, based on a death–birth Markov process, exhibiting an asymptotic power law for $L = 1$. By ignoring the presence of holes, an assumption justified by the fact that their number remains constant and is therefore negligible compared to the asymptotic trends of the corpus, the same approximation can be used for $L > 1$ as well.

We have then confronted our model with the English Wikipedia's growth, which shows initial exponential or large-exponent power-law growth, followed by a slowdown.

We assume that the initial growth was mainly due to the increasing number of users. The subsequent phase was dominated by the filling of gaps and splitting of long pages, a phase still affected by the previous growth trend. The asymptotic phase, which is more similar to our model, should consist in a stable user base and growth dominated by new findings and discoveries, still coupled to the process of filling the gaps.

We have observed some similarities between our model and the latest Wikipedia growth, even if Wikipedia pages are not acyclic (but Wikipedia categories are), and its degree distribution is compatible with a kind of preferential attachment, and therefore, it does not correspond to a fixed connectivity $K$.

Despite this, we found a power law in both trends, and the exponent measured for Wikipedia decreased with time. We think that the present exponent of Wikipedia's trend is due to the fact that it has not yet reached its final stage.

We have shown that it is possible to include a growing number of contributors in our model, and assuming that their number follows exponential or logistic (which shows en early exponential-like behavior) growth, with constant activity per user, it is also possible to approximate the first phases of Wikipedia's growth.

We think that we have provided a constructive agent-based model that reproduces some of the main aspects of Wikipedia's growth. Clearly, a more detailed model of user behavior should be developed and checked against experimental data.

**Author Contributions:** Conceptualization, F.B.; methodology, F.B.; software, F.B. and G.d.B.C.; validation, G.d.B.C.; formal analysis, F.B.; investigation, F.B. and G.d.B.C.; writing—original draft preparation, F.B.; writing—review and editing, F.B. and G.d.B.C.; visualization, G.d.B.C.; supervision, F.B. All authors have read and agreed to the published version of the manuscript.

**Funding:** This research received no external funding.

**Data Availability Statement:** All Wikipedia data used here can be freely downloaded from the Wikimedia Foundation.

**Conflicts of Interest:** The authors declare no conflict of interest. The authors had no role in the collection of data from Wikipedia.

**Sample Availability:** All code is available upon request.

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
