# Peer review of "A Simple Model of Knowledge Scaffolding Applied to Wikipedia Growth"

_futureinternet, doi:10.3390/fi15020067_

Round 1

Reviewer 1 Report

I found this to be an interesting and well-written article, but a technical one. My background is such that I could reasonably follow the technical development. But for many readers I think it would help to clarify a few issues.

1. Define what you mean by an acyclic network, even if it is a standard concept in graph theory.

2. It would help if you can explain better what you mean by scaffolding. In education at least, scaffolding is used as a temporary support for learning, that is later removed (Wood et al.). I’m not sure if it is the best word to choose. (But perhaps it is already used in the literature.)

3. I think some justification is needed for making a linear model. The chain is a chain of articles or knowledge elements, so distances don’t seem to have meaning. Most networks of knowledge I am familiar with, are two-dimensional, where distance does have meaning (e.g., centrality).

4. It might help to explain the origins of the power law behaviours.

5. I was missing a clear articulation of the contribution of the study to knowledge. What do we know now, what can we do now? I understand this is only a first model, and that further development is forthcoming (that also is mentioned in the final sentence).

 Wood, D. et al. "The Role of Tutoring in Problem Solving." Journal of Child Psychology and Psychiatry, vol. 17, no. 2, 1976, pp. 89-100.

Author Response

  1. We added this sentence in the text: In acyclic graphs the nodes are not mutually interlinked with each other therefore the network does not contain any cycles. This means that the graph begins with one node and ends with a different one never returning to a visited node.

  1. Scaffolding here is used in opposition to accumulation that is the act of gathering or amassing, as into a heap or pile. This term has been used in some articles. We modified the phrase adding more context and two citations to literature where scaffolding is used.

  1. We added a part where we explain why we choose an unidimensional linear model.

Since every knowledge bit can be distant from the latest of its prerequisite at maximum distance of L, we could see the distance (d<=L) between nodes as the conceptual distance between the acquisition of one set of knowledge from another, or the background knowledge needed to obtain a certain knowledge. Mathematical theorems are a good example because we have some axioms and with those we could only generate a finite number of theorems (all the ones included in a distance L from the last axiom), but with those theorems we are capable of demonstrate "higher order theorems" that requires more previous knowledge. We added some phrases to make it more clear:

  1. We explained at the end of section for the origins of the power law behaviours.

  1. We added some concepts and some citations in the introduction that should be helpful to better contextualize the study of knowledge and Knowledge Graphs.  Since we are physicists, further development will require contribution of professionals from other fields and new simulations upon which we can base our further analysis, therefore conclusions are not easily determined a priori.

Reviewer 2 Report

Authors propose a simple model of knowledge scaffolding, based on the process of building complex concepts on top of simpler ones (starting from axioms) in an acyclic network shape (limiting the maximum jumps in knowledge). The model allows for measuring its size and time growth, detecting "holes". It is applied to English Wikipedia human-edited articles showing positive results.

The paper focuses an interesting topic, knowledge modelling, and applies it to Wikipedia.

Authors should include more detailed information on the data used. The time period included in the dataset (and the date the information was downloaded from the Mediawiki server). It seems that the model only reflect the grown in the first 2 or 5 months, a really poor contribution.

And that is the main issue of the paper: there are many approaches to Wikipedia grows modelling in scientific journals and even in specific conferences (like OpenSym, or DBpedia ...). There is no "Related work" section were authors should clearly stated what their proposal does that is not done by the many existing ones.

Additionally, there is almost no discussion on the results, and, as a consequence, the conclusions are really vague.

Minor issues: authors claim that "Wikipedia pages are not acyclic", but they are, for example the Article on an actor links to that of a film where he acted and viceversa

In figures suitable axis captions should be included instead of "m,c" and "t" (like in figure 3).

There are some typos in the text: "highest one, AD that will be subsequently filled", ...

Author Response

As suggested we added more specifics on where we retrieved the data and the time period included in the dataset. 

Our purpose was to present a very simple model of knowledge scaffolding we made. Additionally we compared Wikipedia's growth to the obtained trends of our growth to see if our model was at least comparable to an easily accessible knowledge corpus. Our model should describe an approximation of Wikipedia's asymptotic trend as we better explained in the reviewed version of the article. We added a section showing how a growing number of users can be included in the model, in order to better model the actual Wikipedia growth. However, we do not really aim at fitting data, but rather to provide a realistic  agent-based model. 

Our contribution is a general model with only a few variables that roughly model the encyclopedia's trend. We gladly included a citation from OpenSym (the article "The singularity is not near: slowing growth of Wikipedia") and we included an extensive related work section to give more context when we present Wikipedia's growth.

We extended the results chapter.

We stated that Wikipedia pages are not acyclic, you say that they are cyclic, we are stating the same thing . We choose to use a double negative since we used a model that is acyclic and it seemed better to say that Wikipedia is not like our model and therefore not acyclic, than to say that it is cyclic.

The style of figure labels depends on the discipline, In physics we use preferably single-letter, is other fields descriptive labels. We tried to follow the second style, and when it was too long or obvious, we used single letter including their interpretation in the caption.

We corrected the typos we have found, thank you.

Reviewer 3 Report

Advantages.

The topic of this article is very interesting and meaningful.

I enjoyed your article. Ideas is very great!

Disadvantages:

You need to radically revise this article.

- should include an in-depth analysis of the literature on the topic

- place the figures closer to the reference

- caption of figures is so long

- reference [3] is not correct

- you need to number all the formulas

- a more detailed description of the methodology is required.

Author Response

-We added an extensive Related Work section and some other information regarding knowledge and Knowledge Graph studies in the introduction. We also added a section showing the consequences of a growing user base.

Figure placement is an editor's choice.

- Since most readers start by looking at figures, we thick that a self-explanatory caption is more convenient. 

Sorry, reference [3] was a placeholder for a proceedings that was not published yet. We corrected it.

We added the number to all the equations

-We added more information on some topics such as why we chose a linear model, why scaffolding and not accumulation, where the data used come from, how we can interpret some of the variables used,  we also added some literature that might help to give the right context to the article.

Round 2

Reviewer 2 Report

Authors have significantly improved the paper. Anyway, some points need to be fixed before publication:

1 The Introduction includes the introduction of the paper but also a lot of (interesting) information of the background. It should be in a different section

2 The paper should include a explicit Research Question (RQ) that should be included in the introduction, then explain how it is backed by the proposal and finally briefly answered in the cinclusions

3 A "Threat to validity" section should be included, stating actions taken to mitigate them

4 A lot of the text included in the conclusions is not a conclusion itself, but a comment of the results It should be in a "Discussion" section.

Author Response

We have split the Introduction in two, and renamed "Discussion and conclusions" the last section. 

We have included the "explicit research question" and the "threat to validity" in the Introduction, and we have improved the text. 

We limited the changes, since the text was already approved by the other referee, and extensive modification would have implied a further revision. 

Reviewer 3 Report

All is OK

Author Response

Thanks